# Roles of Specialized Chromatin and DNA Structures at Subtelomeres in *Schizosaccharomyces pombe*

**DOI:** 10.3390/biom13050810

**Published:** 2023-05-10

**Authors:** Junko Kanoh

**Affiliations:** Department of Life Sciences, Graduate School of Arts and Sciences, The University of Tokyo, 3-8-1 Komaba, Meguro-ku, Tokyo 153-8902, Japan; jkanoh@bio.c.u-tokyo.ac.jp; Tel.: +81-3-5454-6759

**Keywords:** chromosome end, telomere, subtelomere, shelterin, heterochromatin, knob, gene expression, replication timing, chromatin boundary, evolution, fission yeast

## Abstract

Eukaryotes have linear chromosomes with domains called telomeres at both ends. The telomere DNA consists of a simple tandem repeat sequence, and multiple telomere-binding proteins including the shelterin complex maintain chromosome-end structures and regulate various biological reactions, such as protection of chromosome ends and control of telomere DNA length. On the other hand, subtelomeres, which are located adjacent to telomeres, contain a complex mosaic of multiple common segmental sequences and a variety of gene sequences. This review focused on roles of the subtelomeric chromatin and DNA structures in the fission yeast *Schizosaccharomyces pombe*. The fission yeast subtelomeres form three distinct chromatin structures; one is the shelterin complex, which is localized not only at the telomeres but also at the telomere-proximal regions of subtelomeres to form transcriptionally repressive chromatin structures. The others are heterochromatin and knob, which have repressive effects in gene expression, but the subtelomeres are equipped with a mechanism that prevents these condensed chromatin structures from invading adjacent euchromatin regions. On the other hand, recombination reactions within or near subtelomeric sequences allow chromosomes to be circularized, enabling cells to survive in telomere shortening. Furthermore, DNA structures of the subtelomeres are more variable than other chromosomal regions, which may have contributed to biological diversity and evolution while changing gene expression and chromatin structures.

## 1. Introduction—Telomere and Subtelomere

Chromosomes, which are structures formed by the binding of various proteins and RNA molecules to DNA, serve not only as carriers of genetic information but also as sites for various biological reactions. At the end of eukaryotic linear chromosomes is a region called telomere, where multiple proteins bind to form a complex based on DNA consisting of a tandem repeat sequence [1] (Figure 1). The sequence of telomere DNA differs among species; for example, [TTAGGG]_n_ in vertebrates [2], [TTTAGGG]_n_ in the plant *Arabidopsis thaliana* [3], [TTAGGC]_n_ in the nematode *Caenorhabditis elegans* [4], [TTAC(A)G_2–8_]_n_ in the fission yeast *Schizosaccharomyces pombe* [5], and [T(G)_2–3_(TG)_1–6_]_n_ in the budding yeast *Saccharomyces cerevisiae* [6]. The overall length of telomeric DNA also varies by species, developmental stage, and tissue site, ranging from about 15 kbp in human germ cells to several dozen kbp or more in mice and about 300 bp in fission and budding yeasts [5,6,7,8]. 

The majority of telomeric DNA is double-stranded DNA, with the G (guanine)-rich strand and its complementary C (cytosine)-rich strand. On the other hand, the chromosome ends have a protruding 3′ end, which is called “G-tail” or “G-overhang” (Figure 1). The length of the G-tail is about 150 to 350 bases in human and about 10 to 100 bases in budding yeast [9,10,11]. It is longest during the S (DNA synthesis) phase of the cell cycle, providing a site for telomerase (a reverse transcriptase for telomeric DNA) to act [12]. The telomere elongation by telomerase solves the end replication problem, a challenge that DNA polymerases are unable to fully replicate the ends of linear chromosomes [13,14].

Adjacent to telomeres are regions called subtelomeres, which have characteristic DNA and chromatin structures (Figure 1 and see below). In general, subtelomeres do not contain telomeric tandem repeat sequences (although in rare cases, relatively short telomeric repeat sequences are present within subtelomeres), but, instead, contain common sequences that are highly homologous between subtelomeres of each species. The common sequences are not simple repeats, and it is known in some species that the copy number of multiple common segments (or gene sequences) varies from subtelomere to subtelomere and that they are further combined in complex mosaic-like structures (see below) [15,16,17,18].

The subtelomeres of budding yeast occupy a region roughly 25 kb adjacent to the telomere, with a mosaic of various repeat sequences and variation among chromosome arms [15]. Among these, DNA sequences called X and Y’ elements are found in many subtelomeres [19,20]. The Y’, which is highly homologous across chromosomes, contains ORFs for genes encoding proteins containing RNA helicase-like domains [21]. The X element as a whole has relatively low homology across chromosomes; however, a region called the X core of 475 base pairs is highly conserved across chromosomes and contains the ARS (autonomously replicating sequence) [22,23]. The inner (centromere side) of the X and Y’ elements contain internal repeats with overlapping sequences between several chromosome arms. This region shows variation in length and contains ORFs of genes encoding proteins of the *PAU*, *MAL*, *MEL*, and *ERR* families, or does not contain any ORFs [15].

Human subtelomeres are also a mosaic of various common segments with highly variable sequences [16,17]. Each common segment contains a variety of gene sequences, e.g., the olfactory receptor gene cluster [24] and the *DUX4* gene, which is related to facioscapulohumeral muscular dystrophy [25]. The length of the regions of homology with sequences from other chromosome arms varies with each subtelomere, ranging from less than 20 kb for the shortest to more than 200 kb for the longest [16,17]. Recombination and duplication among human subtelomeres are thought to occur at a higher frequency than in other regions of the chromosome, and diversity is observed even among modern humans [16].

## 2. Subtelomeric DNA in Fission Yeast

The fission yeast *S. pombe* has an exceptionally low number of chromosomes, i.e., only three. Additionally, it stably grows as a haploid, resulting in only six chromosome ends, which makes it highly convenient for studying telomeres and subtelomeres. In the standard wild-type strain 972, there is a subtelomeric region of approximately 100 kb adjacent to telomeres on chromosomes 1 and 2 (Figure 2). Adjacent to the telomeres is a subtelomeric homologous (SH) sequence of approximately 50 kb that shows high homology between the subtelomeres [18,26]. The SH sequence differs from telomere repeat sequences in that it is a complex mosaic of common segmental (or block) sequences of varying sizes [18]. Adjacent to the SH sequence is a subtelomeric unique (SU) sequence of approximately 50 kb. The SU sequences do not show significant homologies between the subtelomeres but are included in the subtelomere because they share a common condensed chromatin structure, a knob (see below for details) [18,27]. Chromosome 3 is distinct from chromosomes 1 and 2 in that the rDNA repeats, which produce rRNAs, are located adjacent to the telomeres at both ends. However, some non-972 strains that are used in laboratories contain a portion (approximately 15 kb) of the SH sequence between the rDNA repeats and both or one of the telomeres on chromosome 3 [26,28]. 

## 3. Fission Yeast Subtelomeres Form Three Distinct Chromatin Structures

### 3.1. Shelterin, a Telomere-Binding Protein Complex

In telomeres, proteins that recognize telomeric repeat sequences bind directly to telomere repeat sequences, which further bind to multiple proteins to form the telomere protein complexes. In organisms such as human and fission yeast, the Pot1 protein that specifically binds to the G-tail at chromosome ends and proteins that specifically bind to the double-stranded telomeric DNA are linked by a group of proteins. This protein complex is called shelterin [29,30]. In fission yeast, the protein Taz1 directly binds to the double-stranded telomere DNA and associates with another protein Rap1 [31,32,33]. In turn, Rap1 further interacts with multiple proteins, such as Poz1. On the other hand, Pot1 associates with Tpz1, which further interacts with Ccq1 and Poz1 [30] (Figure 3). 

The shelterin protects chromosome ends and inhibits the action of DNA repair. Pot1, Tpz1, and Ccq1 are important for protecting telomere ends, and the deletion of either of these proteins causes rapid telomere DNA loss [30,34]. On the other hand, the deletion of Taz1 or Rap1 causes random chromosome end fusion specifically in G1 phase, when Ku-dependent non-homologous end joining (NHEJ) is activated; thus, Taz1 and Rap1 protects telomere end structures by inhibiting NHEJ [35,36,37]. 

The shelterin also play essential roles in maintaining telomeric DNA length. The deletion of Taz1, Rap1, or Poz1 increases telomerase action and results in abnormal elongation of telomere DNA [30,31,32,33]. In contrast, the deletion of Pot1, Tpz1, or Ccq1 not only abolishes the action of telomerase, but also causes telomere DNA to rapidly shorten because the chromosome ends are no longer protected [30,34]. The current model for the shelterin is that when the double-stranded telomeric DNA is long enough, bridge structures frequently form with single-stranded telomeric DNA. These structures may protect the telomeric end structure and inhibit the telomerase action on the telomere termini [30]. Furthermore, the fission yeast shelterin plays important roles in the regulation of chromosome dynamics during mitosis and meiosis and in replication timing control by interacting with other proteins [38,39,40,41,42]. For example, Taz1 associates with Rif1, a global regulator of replication timing, not only at telomeres but also around some late replication origins to regulate replication timing [41,42]. Rif1 is also localized around late replication origins independently of Taz1 by recognizing G-quadruplex structures [43] (Figure 3).

Interestingly, the fission yeast shelterin is localized not only at telomeres but also at the telomere-proximal SH regions spanning approximately 10 kb [44,45] (Figure 2, Figure 3 and Figure 4). However, there are hardly repetitive telomere sequences in the SH regions, and so, the localization mechanism of shelterin remains unclear at present. The deletion of each protein of the shelterin causes derepression of the genes located in the telomere-proximal SH region; thus, the shelterin itself possibly forms a repressive chromatin structure at subtelomeres as well as at telomeres [45]. On the other hand, the shelterin at SH regions is able to establish heterochromatin by recruiting Clr4, a histone H3K9 methyltransferase [44,46] (Figure 4 and see below).

### 3.2. Subtelomeric Heterochromatin

Fission yeast chromosomes form constitutive heterochromatin at pericentromeres, subtelomeres, and the mating-type *mat* locus. Heterochromatin at the pericentromeres contributes to accurate chromosome segregation [47], and heterochromatin at the *mat* locus suppresses the function of one of the two zygotic types (*h^−^* and *h*^+^), thus suppressing gene expression through the condensed structure of heterochromatin [48]. Heterochromatin at subtelomeres also has a repressive effect on gene expression [44]; however, the details of its cellular function remain unknown. 

In the heterochromatin of fission yeast, the histone methyltransferase Clr4 is first recruited to a defined region and methylates histone H3K9, and HP1 (heterochromatin protein 1) family proteins bind to the H3K9me (histone H3 methylated at K9) [46]. In turn, HP1 further recruits Clr4, which induces methylation of nearby H3K9 [46]. In this way, the localization of HP1 spreads, and adjacent HP1 proteins bind to each other to form a condensed chromatin structure [49].

At the subtelomeres, heterochromatin formation is initiated by two independent pathways. The first pathway involves the shelterin proteins Taz1 and Ccq1, which recruit Clr4 and establish heterochromatin at the telomere-proximal subtelomere regions [44,50] (Figure 4). The second pathway involves RNAi (RNA interference) [44,51]. The subtelomeres and *mat* locus contain sequences that are homologous to part of the duplicated sequences (dg and dh repeats) found in the pericentromeres [44,52]. Interestingly, the SH sequences of the subtelomeres of chromosomes 1 and 2 in the strain 972 contain the RecQ-type DNA helicase genes *tlh1-4*, respectively [18], which contain part of the dh sequence within their coding regions [44]. RNAi acts on the dh sequence in the transcripts from the *tlh* genes and produces siRNA (small interfering RNA), which recruits Clr4 to the dh region of the subtelomeres (the *tlh* gene loci) to establish heterochromatin (Figure 4). This RNAi-mediated establishment of heterochromatin is common among constitutive heterochromatin [53,54], and it can be speculated that originally a single sequence is now used in three chromosomal domains. The established heterochromatin spreads over the SH regions [26,44] (Figure 2 and Figure 4). Thus, SH sequences play important roles in subtelomeric heterochromatin formation.

### 3.3. Another Condensed Chromatin Structure, a Knob, in SU Regions

As mentioned above, subtelomeres in fission yeast consist of SH and SU regions (Figure 2). Despite low homology between DNA sequences in the SU regions, they form a common condensed chromatin structure, a knob, which is intensively stained by DAPI (4′,6-diamidino-2-phenylindole) specifically in interphase cells [55] (Figure 2). Knob, unlike heterochromatin, does not show H3K9me and almost no localization of HP1 proteins [27,55]. Interestingly, however, the Sgo2 protein, which localizes to the centromeres in M phase for precise chromosome segregation [56,57], is recruited throughout the subtelomeres during interphase and plays an essential role in the knob formation [27].

The SU sequences of the strain 972 contain various genes that are involved in nutrient metabolism and stress responses. Knob has the effect of suppressing gene expression, although not to the same extent as heterochromatin, so that genes in the SU region, such as galactose metabolism genes (*gal1*^+^, *gal7*^+^, and *gal10*^+^), are mildly inhibited [27]. For this reason, fission yeast, unlike budding yeast, hardly grows on galactose media [58]. Thus, it can be speculated that genome organization occurred during evolution, and fission yeast survived only in an environment in which galactose is not required. In other words, by examining the functions of the gene clusters present in the SU sequences of fission yeast, we can infer the environment in which fission yeast survived during the evolutionary process.

Intriguingly, Sgo2 plays another role in subtelomeres. The deletion of Sgo2 results in the abnormal acceleration of DNA replication only at late replication origins in the subtelomeres; thus, Sgo2, possibly knob formation, inhibits initiation of DNA replication until late S phase [27]. However, the mechanisms underlying this regulation are currently unknown. 

### 3.4. Boundary between the Subtelomeres and Euchromatin

In the *SD5* (subtelomere deletion 5) strain, in which all the five SH regions were completely deleted from the 972-derived strain (note that this strain also has an SH sequence on the left arm of chromosome 3, i.e., it has five SH sequences), RNAi cannot act on SH regions; however, heterochromatin is established by the shelterin proteins at the chromosome ends and spreads into the SU region, although it does not spread out from the SU region (Sgo2 localization region) [26] (Figure 5). Even if approximately half of the SU sequence is further deleted, neither heterochromatin nor Sgo2 extends its localization beyond the original SU region [26] (Figure 5). This suggests that there is a boundary mechanism between the SU region and the euchromatin region that prevents heterochromatin and knob from entering the euchromatin. If such mechanism does not exist there, heterochromatin would disorderly spread into the euchromatin region, repressing gene expression and threatening cell viability. Interestingly, in the region that seems to be the boundary, nucleosome structures are missing [26]. Therefore, it is possible that non-histone proteins are tightly bound to block chromatin outflow, or that the lack of histones blocks the spread of heterochromatin (Figure 5).

## 4. Roles of Subtelomere DNA Sequences in Fission Yeast

### 4.1. Role of the SH Sequences in Protecting Expression of the Genes in SU from Heterochromatin Spreading

The strain *SD5* shows normal cell proliferation and cell morphology in a nutrient-rich medium [26]. Furthermore, *SD5* cells do not have defects in responses to various forms of stress (high and low temperatures, DNA damages, and destabilization of microtubules), in meiosis and spore formation after nitrogen starvation, and in telomere length control [26]. However, they exhibit marked sensitivity to high concentration of Na^+^ and Li^+^ due to repression of the *nhe1* gene encoding a plasma membrane Na^+^ (or Li^+^)/H^+^ antiporter by heterochromatin invasion of the SU region. In fact, heterochromatin spread into all of the SU regions in chromosomes 1 and 2 [26] (Figure 5). Thus, the SH region serves as a buffer zone against heterochromatin spreading.

### 4.2. Roles of Subtelomere DNA in Cell Survival in Telomere Shortening

#### 4.2.1. Roles of SH Sequences in Self-Circularization of Chromosomes

When genes such as Pot1, which are important for chromosome end protection, are deleted, telomere DNA rapidly shortens, resulting in random interchromosomal end fusion and fatal chromosome segregation, and most cells die [34]. Some cells, however, are able to survive. Interestingly, in most of those survivors, all three chromosomes are stabilized by self-circularization. In other words, they have ring-shaped chromosomes just like those in prokaryotes [34] (Figure 6). How is this possible? Interestingly, the SH sequences are equipped with five common sequence motifs that are oriented in the same direction when circularized [59]. Using either set of these sequences on chromosomes 1 and 2, all of the three chromosomes in the surviving cells became self-circularized via single-strand annealing (SSA) (note that the exact mechanism of chromosome end fusion in the rDNA repeat regions of chromosome 3 is not yet known) [59]. Thus, SH sequences play an important role in promoting self-circularization of chromosomes, i.e., in enabling cells to survive the crisis of telomere shortening. This is a feat made possible by the fact that fission yeast has only three chromosomes. That is to say, chromosome self-circularization, in which the ends of each chromosome fuse with each other without fusing with different chromosome ends, would be almost impossible in species with larger number of chromosomes. Similarly, when telomerase is inactivated in the presence of Pot1, telomeric DNA is gradually shortened with each cell division, and most cells die. However, in this case, a small fraction of the survivors maintains linear state of their chromosomes by amplification or recombination of telomeres, SH or rDNA, in addition to self-circularization of chromosomes [60,61] (Figure 6).

#### 4.2.2. Roles of Non-SH Sequences in Circularization of Chromosomes in the Absence of SH Sequences

Then, are SH sequences essential for fission yeast to survive telomere shortening? Surprisingly, even when telomere shorteining is induced in the *SD5* strain, the strain survives by means that do not rely on recombination with SH sequences. How is this possible? There are multiple sequences other than SH sequences that are in the same orientation when chromosomes are circularized, such as retrotransposon LTR (long terminal repeat) and *L-asparaginase* gene sequences, within or near SU regions, and the chromosomes are circularized by the SSA reaction at these common sequences [26]. Thus, there are multiple defenses against telomere shortening at and near subtelomeres in fission yeast.

More surprisingly, in the absence of SH sequences, when telomere shortening is induced, not only do cells with each of the three chromosomes self-circularize, but also survivors emerge where chromosomes 1 and 2 fused at the LTR and other common sequences to form a single circularized chromosome [26]. How do they survive with two centromere sequences in one circularized chromosome (i.e., dicentric chromosome)? The chromosome is maintained as stable by the inactivation of one of the two centromeres [26]. In these cells, the centromere sequence itself is not lost, but the localization of CENP-A (a histone H3 variant), which is the basis for kinetochore formation in the core region of the centromere [62], is lost from nucleosomes at the core centromere, and normal histone H3 is incorporated in its place and its K9 are methylated to form heterochromatin [26]. How inactivation of one centromere occurs and how this inactivation is maintained is not yet clear at present.

## 5. Perspective

Subtelomeres are equipped with multiple layers of defense mechanisms to inherit chromosomes to the next generation, i.e., to sustain life. In doing so, subtelomeres may undergo dynamic structural changes, such as the transformation of linear chromosomes into circularized chromosomes. In fact, it was recently shown that subtelomeres are more prone to structural changes than other chromosomal regions even without exposure to critical situations, i.e., subtelomeres are hotspots of genome evolution [18]. Interestingly, *S. pombe* natural wild strains isolated in various countries show significant variations in the DNA structure of subtelomeres. Some of them lack genes present in the SU region of 972 or, conversely, increased their copy number [18]. Furthermore, it is highly likely that the large-scale changes in subtelomeric sequences resulted in changes in the distribution of heterochromatin and knob, which have the effect of suppressing gene expression. In summary, even in the same *S. pombe*, differences in gene expression around subtelomeres would result in differences in a variety of cellular functions.

The reason why subtelomere changes are so striking is probably that subtelomeres often have overlapping genes because they share common sequences, and even if one gene is mutated or deleted, it is not likely to be life-threatening, i.e., not lethal. This may be because subtelomeric changes are easily inherited to the next generation. Conversely, in the case of genes that have only one copy inside the chromosome, the accumulation of mutations may be life-threatening at a faster rate and less likely to remain in future generations.

In humans as well as in fission yeast, subtelomeres are hotspots of genomic variation [16]. Since the human subtelomeric common sequences contain a variety of genes, differences in subtelomere sequences would be linked to human diversity. In addition, in chimpanzees, bonobos and gorillas, which are considered to be evolutionarily most closely related to humans, there is a domain between the telomere and subtelomere common sequence that consists of a 32 bp unit repeat sequence called the subterminal satellite (StSat), which is completely absent in humans [63]. It is interesting to see how the presence or absence of this StSat sequence affected the chromatin structure of neighboring subtelomeres, gene expression, and subtelomere diversity. In summary, while chromosome terminal regions play an important role in maintaining chromosomes from various aspects, subtelomeres themselves dynamically changed their structures over the long history of living organisms. This may have altered gene expression around subtelomeres and contributed to the diversity and evolution of organisms.

## Figures and Tables

**Figure 1 biomolecules-13-00810-f001:**
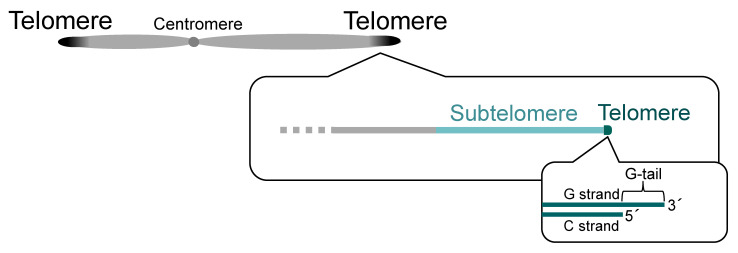
**Telomeres and subtelomeres.** Telomeres exist at the ends of a linear chromosome. Most telomeric DNA is double-stranded, but single-stranded telomeric DNA (G-tail) is present at the most terminal portion of the chromosome. Subtelomeres are regions adjacent to telomeres.

**Figure 2 biomolecules-13-00810-f002:**
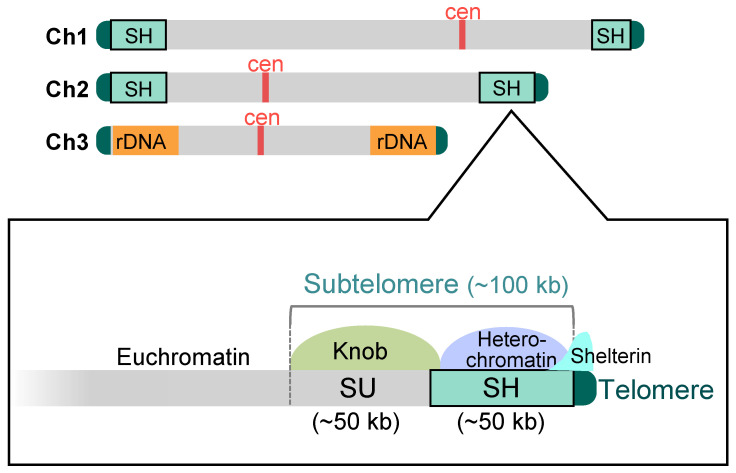
**Schematic structure of the subtelomeres in fission yeast.** In the standard wild-type strain 972, there exist four subtelomere regions in chromosomes 1 and 2 (approximately 100 kb). The subtelomere consists of two distinct regions, SH and SU. The SH region (approximately 50 kb) possesses common DNA sequences homologous with other subtelomeres and forms heterochromatin. On the other hand, the SU regions (approximately 50 kb) do not show high homology with each other, whereas they share a condensed chromatin structure, a knob. The shelterin complex is localized at the telomere-proximal SH region as well as at telomeres. In chromosome 3, rDNA repeat sequences are located adjacent to the telomeres. Ch1–3: chromosomes 1–3; cen: centromere.

**Figure 3 biomolecules-13-00810-f003:**
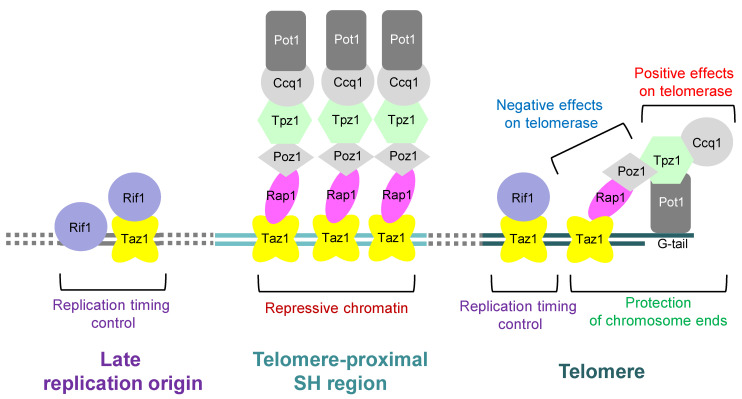
**Roles of the telomere-binding proteins in fission yeast.** In fission yeast, the shelterin complex consists of Pot1, Tpz1, Ccq1, Poz1, Rap1, and Taz1, and protects chromosome ends. Furthermore, Pot1-Tpz1-Ccq1 plays a role in promoting the action of telomerase on telomeres, whereas Poz1-Rap1-Taz1 represses the action of telomerase. The shelterin complex is also localized at the telomere-proximal SH regions to form repressive chromatin. Moreover, Taz1 interacts with Rif1 to regulate replication timing not only at telomere but also around some internal late replication origins. Note that Rif1 is also localized around late replication origins independently of Taz1.

**Figure 4 biomolecules-13-00810-f004:**
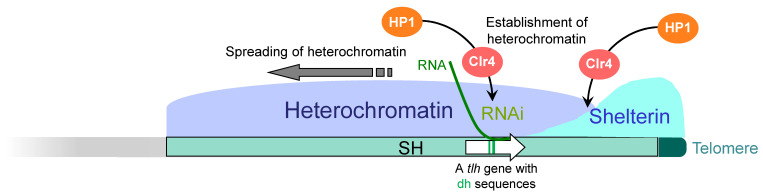
**Mechanism of subtelomeric heterochromatin formation in fission yeast.** Shelterin, which binds to the telomere, is also localized to the telomere-proximal SH region by a currently uncharacterized mechanism. Taz1 and Ccq1 in the shelterin recruit the histone H3K9 methyltransferase Clr4. In addition, the RNAi machinery acts on the RNA molecules transcribed from a DNA helicase gene (*tlh*) located in the SH region and recruits Clr4 independently of the shelterin. Both of them induce methylation of histone H3K9, which recruits HP1 proteins (establishment of heterochromatin). Subsequently, HP1 recruits Clr4, which in turn methylates nearby histone H3K9, and HP1 binds to it. In this way, the heterochromatin spreads over the SH region.

**Figure 5 biomolecules-13-00810-f005:**
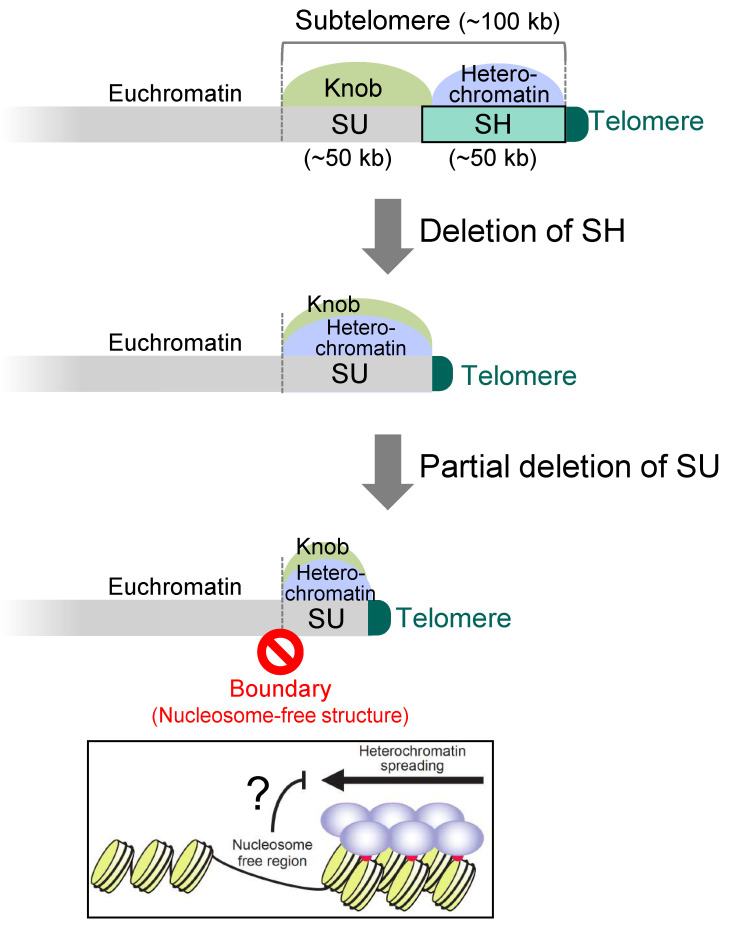
**Chromatin boundary between the subtelomere and its adjacent euchromatin.** After the complete deletion of SH, heterochromatin invades the whole SU region. Furthermore, even after the partial deletion of SU, heterochromatin and knob (Sgo2) do not spread to the adjacent euchromatin, suggesting the existence of a chromatin boundary mechanism between subtelomeres (SU) and the euchromatin to protect gene expression at euchromatin. The absence of nucleosome structures itself or unidentified proteins associated at the putative boundary may inhibit the heterochromatin boundary.

**Figure 6 biomolecules-13-00810-f006:**
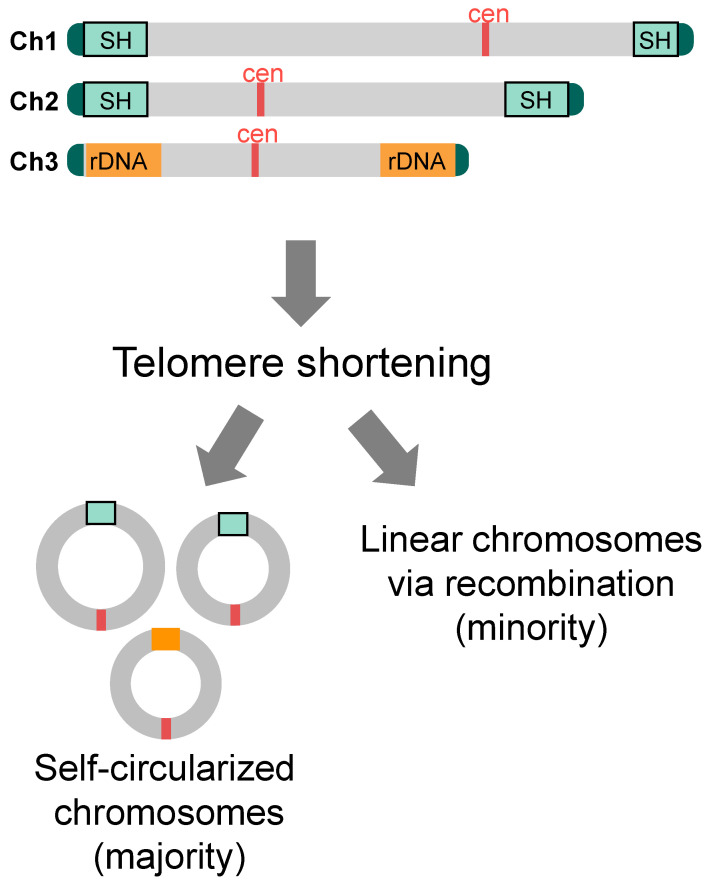
**Cell survival after telomere shortening.** When telomere shortening is induced by Pot1 deletion in cells with SH sequences, all of the survivors possess self-circularized chromosomes utilizing common sequence motifs within SH and possibly rDNA sequences for the SSA reaction. If telomere shortening is induced by inactivation of the telomerase, minority of survivors emerges that possesses linear chromosomes that are maintained by recombination at telomeres, subtelomeres, and/or rDNA repeats.

## Data Availability

Not applicable.

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
