# Peer review of "Roles of Specialized Chromatin and DNA Structures at Subtelomeres in Schizosaccharomyces pombe"

_biomolecules, 2023, doi:10.3390/biom13050810_

Round 1
Reviewer 1 Report
Dear Editor and author,
I appreciate the invitation to review the manuscript “Multistep chromatin regulation at chromosome ends”. In this review, the authors present the information about telomere and subtelomere of Schizosaccharomyces pombe showing its importance in different cellular mechanisms. The organization of the text is very clear and all topics are well discussed based on the literature. I consider that the manuscript can be accepted with minor corrections.
I put my considerations as comments in the text below
1. Add the name of the species (Schizosaccharomyces pombe) to the title, since most of the text deals with data referring to this species: “Multistep chromatin regulation at chromosome ends of Schizosaccharomyces pombe.
2. At several points in the text I suggest that the author mention the articles in which the information was taken.
Ex. Topic 4, line 3: “The DNA sequences of subtelomeres vary among species…”
Topic 4, line 6: “The subtelomeres of budding yeast…”
3. At the beginning of the introduction, when the author comments on the sequences of some telomeres, I suggest replacing the reference [2], from Drosophila, by other more specific ones
4. I suggest changing the title of subtopic 3.4. Telomere and membrane dissociation may be important for correct segregation, which makes perfect sense. However, cycle regulation is done by cyclin/CDK and not by telomere action.
Author Response
Please see attachment, thanks.

Reviewer 2 Report
This is a nice and interesting review on telomeric and subtelomeric functions focusing on fission yeast. The title does not imply that this work mostly covers fission yeast results. It would be thus appropriate to adjust the title or to expand on telomeric and subtelomeric functions in other organisms. Personally, I would prefer the latter option and suggest to expand on the implications for vertebrate/human telomers, which is only shortly mentioned in the final paragraph. Alternatively, a broader perspective including other yeast species (including budding yeast) might be considered.
In general, the manuscript would however benefit from language editing. Many terms could be phrased simpler by a professional editing service. This would also help making this piece attractive for a larger audience.
Minor point:
Fig. 5: the upper “cen” labeling is not readable.
Author Response
Please see the attachment, thanks.

Reviewer 3 Report
In this review, the author provides an overview of the chromatin biology at the end of the chromosome including the telomere and subtelomere. The review includes information about telomeric DNA, the shelterin complex, subtelomeres, and the epigenetic state of these regions.
This is an important topic that will be of interest to the genome maintenance and epigenetics fields. However, this review has some important limitations that should be addressed. The manuscript has an uneven amount of details in the different sections. While the title of the review suggests a broad review that includes a description of telomeres in eukaryotes, the manuscript instead focuses heavily on subtelomeres primarily in the yeast S. pombe. Additionally some key telomere processes are insufficiently described, including telomerase.
Major points
- The title of the review suggests a comprehensive review of chromatin biology of telomeres in eukaryotes. However, there are rather sparse details about many aspects of telomeres compared to the extensive descriptions of subtelomeres. Additionally, the manuscript focuses heavily on S. pombe which appears to have telomere/subtelomere processes that are not applicable to all eukaryotes (such as chromosome circularization). I suggest the author either 1) edit the title and manuscript to make the review more focused on subtelomeres in yeast, or 2) provide additional information about telomeres (outlined below) as well as additional comparisons to other eukaryotes.
- The description of telomeres is lacking sufficient detail. I suggest that the author provide additional information about several key processes including telomerase, T-loops, TERRA, and the end replication problem.
- Section 2 lacks details about the shelterin components making it hard to follow the points in section 3 and figure 2. I suggest that this section is edited to include details of each shelterin component, including their binding site/partners and function.
Minor points
- Section 3.1 does not include any details about how the shelterin complex limits DNA repair processes.
- Section 3.3 and 3.4 are fairly redundant and difficult to read in their current order. I suggest that this is edited to first describe the role of telomeres in mitosis and then include any additional meiosis specific roles.
- While the subtelomeric chromatin/epigenetic state is described in detail, this same type of description is not provided for telomeres.
- Section 6 describes the role of subtelomeres in cell survival after telomere loss, however the review lacks a more comprehensive description of the causes/consequences of telomere loss.
Author Response
Responses to the reviewers’ comments (RE: biomolecules-2309414)
I greatly thank the reviewer for valuable suggestions. I have extensively modified my manuscript according to the reviewer’s comments.
This is an important topic that will be of interest to the genome maintenance and epigenetics fields. However, this review has some important limitations that should be addressed. The manuscript has an uneven amount of details in the different sections. While the title of the review suggests a broad review that includes a description of telomeres in eukaryotes, the manuscript instead focuses heavily on subtelomeres primarily in the yeast S. pombe. Additionally some key telomere processes are insufficiently described, including telomerase.
- The title of the review suggests a comprehensive review of chromatin biology of telomeres in eukaryotes. However, there are rather sparse details about many aspects of telomeres compared to the extensive descriptions of subtelomeres. Additionally, the manuscript focuses heavily on S. pombe which appears to have telomere/subtelomere processes that are not applicable to all eukaryotes (such as chromosome circularization). I suggest the author either 1) edit the title and manuscript to make the review more focused on subtelomeres in yeast, or 2) provide additional information about telomeres (outlined below) as well as additional comparisons to other eukaryotes.
>> I thank the reviewer for pointing out the imbalance in the content. Based on the reviewer's comments, I have removed most of the sections on telomeres from the manuscript. As a result, the current manuscript now focuses on the subtelomeres in fission yeast. Accordingly, I have changed the title to fit the content of the current manuscript.
- The description of telomeres is lacking sufficient detail. I suggest that the author provide additional information about several key processes including telomerase, T-loops, TERRA, and the end replication problem.
>> As described above, I have removed most of the sections on telomeres from the manuscript. Instead of going into detail of the basics of telomeres and blurring the focus, I decided to concentrate on the subtelomeres in fission yeast in this manuscript, but I mentioned telomerase in the original manuscript and have added brief explanations of the basics of telomeres in the current manuscript.
- Section 2 lacks details about the shelterin components making it hard to follow the points in section 3 and figure 2. I suggest that this section is edited to include details of each shelterin component, including their binding site/partners and function.
>> Since the current manuscript focuses on subtelomeres, I would not like to discuss shelterin in detail, such as their binding sites. Instead, I have added a brief introduction for each protein and a summary of the roles of shelterin. I have also added the explanation for Rif1.
- Section 3.1 does not include any details about how the shelterin complex limits DNA repair processes.
>> I thank the reviewer for pointing out the missing information in the manuscript regarding the role of shelterin in inhibiting DNA repair.
- Section 3.3 and 3.4 are fairly redundant and difficult to read in their current order. I suggest that this is edited to first describe the role of telomeres in mitosis and then include any additional meiosis specific roles.
>> I thank the reviewer for the suggestion. As described above, I have removed those sections from the manuscript.
- While the subtelomeric chromatin/epigenetic state is described in detail, this same type of description is not provided for telomeres.
>> I thank the reviewer for the suggestion. As described above, I have removed most of the sections on telomeres from the manuscript. Instead, the subtelomeric gene silencing by shelterin is described in the current manuscript.
- Section 6 describes the role of subtelomeres in cell survival after telomere loss, however the review lacks a more comprehensive description of the causes/consequences of telomere loss.
>> I have added some explanation regarding fatal chromosome segregation. Please note that the explanation about the survivors by recombination has been already described in the original manuscript.
Round 2
Reviewer 3 Report
The author has adequately addressed all my comments.
Author Response
Thanks.